# Severe Secondary Atrophic Rhinitis with Extensive Osteomyelitis Following COVID-19-Associated Necrotizing Rhinitis: A Case Report and Microbiological Analysis

**DOI:** 10.3390/reports8040237

**Published:** 2025-11-18

**Authors:** Anton Danylevych, Sofiya Tsolko, Iryna Tymechko, Olena Korniychuk, Yulian Konechnyi

**Affiliations:** 1Lviv Territorial Medical Union «Multidisciplinary Clinical Hospital of Emergency and Intensive Care», Mykolaichuka St., 9, 79059 Lviv, Lviv Region, Ukraine; adanylevych@gmail.com (A.D.); sofija.lviv@gmail.com (S.T.); 2Dolishniy Institute of Regional Research, National Academy of Sciences of Ukraine, Kozelnytska St., 4, 79034 Lviv, Lviv Region, Ukraine; tymi@ukr.net; 3Department of Microbiology, Danylo Halytsky Lviv National Medical University, Zelena St., 12, 79010 Lviv, Lviv Region, Ukraine; o_korniychuk@ukr.net

**Keywords:** atrophic rhinitis, ozena, COVID-19, necrotizing rhinitis, *Pseudomonas aeruginosa*, *Staphylococcus aureus*, empty nose syndrome, Antimicrobial Resistance

## Abstract

**Background and Clinical Significance**: Atrophic rhinitis (AR) is a rare, chronic inflammatory condition characterized by progressive atrophy of the nasal mucosa and underlying bone. The present report describes a case of severe secondary AR as a sequela of COVID-19-associated necrotizing rhinitis, highlighting the diagnostic and management challenges posed by multi-drug resistant pathogens and extensive anatomical destruction. **Case Presentation:** A 75-year-old female developed progressive necrotizing rhinosinusitis with osteomyelitis following a COVID-19 infection. Computed tomography (CT) confirmed an osteolytic process and subsequent profound anatomical destruction, while histopathology ruled out invasive fungal disease. The resulting cavity was colonized by multi-drug resistant *Pseudomonas aeruginosa* and *Staphylococcus aureus*. **Management and Outcome:** Management focused on preventing crust formation through a structured “nasal rest” protocol, supplemented by cleansing nasal douching with a surfactant (baby soap) and mechanical crust removal. This treatment led to significant clinical improvement, with reduced crusting and complete resolution of ozena symptoms. **Conclusions**: This case illustrates the potential for SARS-CoV-2 infection to precipitate severe necrotizing sinonasal complications leading to secondary AR. It demonstrates the efficacy of a management strategy focused on mechanical cleansing and nasal rest, particularly when conventional antibiotic therapy is limited by extensive drug resistance.

## 1. Introduction and Clinical Significance

Atrophic rhinitis (AR) is an uncommon, chronic, and progressive inflammatory disease affecting the nasal mucosa. It is defined by the atrophy of the mucosal lining and the underlying bone of the nasal turbinates, which leads to a paradoxically widened nasal cavity, the formation of thick, dry crusts, and a characteristic foul odor known as ozena [1]. The condition is broadly classified into two forms. Primary AR is idiopathic, more prevalent in resource-limited countries, and often associated with specific pathogens like *Klebsiella ozaenae* [2,3,4]. In contrast, secondary AR, which is more frequently encountered in resource-abundant nations, develops as a consequence of known preceding events such as extensive sinonasal surgery, trauma, radiation therapy, or chronic granulomatous diseases like sarcoidosis or tuberculosis [4,5,6].

The pathophysiology of AR involves a metaplastic transformation of the normal ciliated pseudostratified columnar epithelium into a non-functional, stratified squamous epithelium. This is accompanied by a significant loss of cilia, mucus-producing goblet cells, and submucosal seromucinous glands [7,8]. The resulting impairment of mucociliary clearance creates a stagnant environment that is highly susceptible to chronic bacterial colonization and infection, perpetuating a vicious cycle of inflammation, crust formation, and tissue damage [9,10].

The clinical impact of AR extends beyond physical symptoms. While patients suffer from chronic nasal obstruction, dryness, and discomfort, the psychosocial burden is often more debilitating. The foul odor, typically imperceptible to the patient due to associated anosmia, can lead to significant social stigma, embarrassment, and isolation, contributing to anxiety and depression [11,12,13].

This case report aims to describe a particularly severe presentation of secondary AR that developed as a sequela of a COVID-19-associated necrotizing infection. We detail the complex polymicrobial bacteriology, characterized by multi-drug resistant organisms, and present the successful outcome of a management strategy centered on aggressive mechanical debridement and biofilm disruption, offering valuable insights into the treatment of this challenging condition.

## 2. Case Presentation

### 2.1. Patient Background and Clinical History

A 75-year-old female was evaluated in 2024 during hospitalization for pneumonia and pulmonary embolism. An otolaryngology consultation was requested due to severe and worsening nasal symptoms, including complete nasal obstruction, extensive crusting, and a potent foul odour.

Her relevant medical history began in October 2021 with a SARS-CoV-2 infection, which was confirmed by a rapid antigen test and treated conservatively. Approximately two weeks after the diagnosis, her nasal symptoms began to worsen, with the onset of nasal congestion, purulent discharge, and a foul offensive odour that persisted thereafter. The patient did not seek medical attention and performed only occasional nasal irrigation with a syringe. In August 2022, due to a significant worsening of her condition, including the new onset of eye pain and complete nasal blockage, she was admitted to the hospital for the first time.

Following an initial assessment in late August 2022, a partial debridement was performed, and tissue was sent for analysis. The first histopathological examination revealed fragments of bone trabeculae amidst extensive purulent-necrotic debris, with areas of focal inflammatory cell infiltration and associated haemorrhages. A course of conservative treatment was initiated, including systemic antibiotics, anti-inflammatory drugs, intensive nasal irrigation, and topical antibiotics. However, due to a lack of clinical improvement, a second, more targeted surgical debridement (necrosectomy of the posterior parts of the nasal septum and anterior ethmoid cells bilaterally) was performed in mid-September 2022. Despite this second intervention, the patient’s condition remained unchanged.

A computed tomography (CT) scan of the paranasal sinuses from this period revealed an osteolytic process (bone destruction) of the ethmoid labyrinth cells, with extension to the ostium of the right frontal sinus and to the left infundibulum of the maxillary sinus. No signs of extension through the lamina papyracea to the orbits were found. Given the progressive nature of the disease, a decision was made to proceed with a radical surgical intervention to remove all altered tissue. In late September 2022, the patient underwent a major polysinusotomy with necrosectomy, during which the maxillary, anterior and posterior ethmoid, frontal, and sphenoid sinuses were opened and intensively irrigated. Intact mucosa was spared where possible. The final histopathological examination from this surgery confirmed the previous findings, describing: (1) areas of focally necrotic tissue with associated haemorrhages, and (2) fragments of bone trabeculae surrounded by extensive purulent-necrotic debris, accompanied by multiple small-focal areas of inflammatory cell infiltration and haemorrhages. Bacteriological examination during early September 2022 identified: *Proteus vulgaris* (10^4^ CFU/mL), and *Enterococcus* sp. (10^3^ CFU/mL). During early October 2022 *Candida* spp. (10^4^ CFU/mL) was identified. Neither PCR, nor fungal culture were performed during or after treatment.

Based on the combination of the clinical presentation, radiological findings, and histopathological results, a final diagnosis of necrotizing rhinosinusitis with osteomyelitis of the midfacial skeleton was established. Invasive fungal sinusitis was ruled out due to the absence of characteristic fungal elements on both histopathology examinations. Although serological testing for granulomatosis with polyangiitis (GPA) was not performed, this diagnosis was considered unlikely based on the localized clinical presentation, non-specific histopathology, and the subsequent disease progression.

Following the major surgery, treatment remained conservative, consisting of mechanical removal of crusts, discharge, and minor necrotic elements. Over time, as a result of the persistent inflammation, extensive mucosal damage, and altered airflow within the enlarged nasal cavity, the condition evolved into severe secondary atrophic rhinitis. This led to further degradation of the nasal structures, with eventual resorption of the inferior and middle conchae and the lateral nasal walls, creating the single, enlarged nasal cavity observed at the 2024 evaluation. Short table with timeline of clinical evolution is presented in Table 1.

### 2.2. Clinical and Radiological Findings (2024)

On physical examination, the patient’s nasal cavities were entirely filled with thick, green-brown crusts, which completely obstructed both the anterior and posterior nasal airways, causing marked respiratory difficulty. An intense foul odor (ozena) was present.

Endoscopic examination after initial debridement revealed an extremely widened nasal cavity with a complete absence of key anatomical structures. The inferior, middle, and superior turbinates were not identifiable. A large perforation was noted in the posterior nasal septum. The lateral nasal walls were also absent, creating a common cavity that exposed the lateral walls of the maxillary sinuses bilaterally. Posteriorly, two distinct openings were visible, which were determined to be the ostia of the sphenoid sinuses. Superiorly, remnants of the frontal recess were identified, and the frontal sinus appeared intact. The progression of healing is documented in Figure 1 and Figure 2.

Computed tomography (CT) scans of the paranasal sinuses confirmed the extensive structural destruction consistent with severe atrophic rhinitis and advanced osteomyelitis (Figure 3). The images showed a markedly widened nasal cavity with complete loss of the nasal turbinates and a large septal perforation. The lateral nasal walls were severely eroded, resulting in direct communication between the nasal cavity and the maxillary sinuses. The ethmoid air cells were almost entirely absent. Diffuse mucosal thickening was observed throughout the remaining sinonasal cavity, indicating chronic inflammation.

### 2.3. Microbiological Investigations and Review

A literature review was conducted using PubMed, Google Scholar, and Scopus databases. During the preparation of this manuscript, the generative artificial intelligence (AI) tool Google Gemini 2.5 Pro was utilized. Its application was strictly limited to two specific tasks: (1) improving the grammar, syntax, and readability of the text (language editing), and (2) assisting in the formulation of search queries for literature databases such as PubMed and Google Scholar (literature search assistance). The AI tool did not contribute to the generation, analysis, or interpretation of any clinical data. Furthermore, it was not used to generate any of the scientific analysis, discussion points, or conclusions presented in this paper. The authors assume full and complete responsibility for all content, including the final wording and scientific assertions, of the manuscript.

#### 2.3.1. Sample Collection, Culture, and Identification

Nasal swab specimens were collected from the crusted mucosal surfaces of the nasal cavity three months after the initiation of treatment. The swab was immediately eluted into a standardized 1.0 mL volume of sterile saline/transport buffer, and were immediately transported to the microbiological laboratory of the Municipal Non-Profit Enterprise Lviv Territorial Medical Union «Multidisciplinary Clinical Hospital of Emergency and Intensive Care» in Lviv, Ukraine.

For bacterial isolation, the specimens were inoculated onto primary culture media, including 5% Sheep Blood Agar to support the growth of both fastidious and non-fastidious organisms, and MacConkey Agar for the selective isolation and differentiation of Gram-negative bacilli. Serial dilutions were prepared from this liquid eluate, and 0.1 mL was plated to obtain the CFU/mL concentration. The plates were incubated under aerobic conditions at 35–37 °C for 24–48 h. Bacterial identification was performed based on colony morphology, Gram stain characteristics, and a panel of standard biochemical tests. *P. aeruginosa* identity was confirmed by positive Oxidase test, non-fermentative metabolism on TSI, *S. aureus* was confirmed by positive Catalase test and Coagulase test (tube method).

The isolates were identified as *Pseudomonas aeruginosa* and *Staphylococcus aureus*. Quantitative analysis revealed a concentration of 10^5^ Colony Forming Units per milliliter (CFU/mL) for *P. aeruginosa* and 10^6^ CFU/mL for *S. aureus*.

The presence of a resilient, polymicrobial biofilm was strongly inferred based on a confluence of clinical and microbiological evidence, rather than direct laboratory visualization. This inference is supported by several key factors: (1) the well-established and potent biofilm-forming capabilities of both *P. aeruginosa* and *S. aureus*, which are extensively documented as key pathogens in chronic, persistent infections 1; (2) the clinical presentation of thick, tenacious, and malodorous crusts that were physically difficult to remove, consistent with a matrix-encased microbial community; (3) the persistence of the infection and high bacterial load despite initial surgical debridement; and (4) the extreme multi-drug resistance profile of the P. aeruginosa isolate, a phenotype that is frequently associated with the protected environment of a biofilm community.

#### 2.3.2. Antimicrobial Susceptibility Testing (AST)

Antimicrobial susceptibility testing was performed using the Kirby-Bauer disk diffusion method, with procedures and interpretation carried out in strict accordance with the European Committee on Antimicrobial Susceptibility Testing (EUCAST) guidelines, version 14.0 (2024) with relevant quality control [2].

An inoculum for each isolate was prepared by suspending colonies in sterile saline to achieve a turbidity equivalent to the 0.5 McFarland standard, corresponding to an approximate bacterial concentration of 1–2 × 10^8^ CFU/mL. The entire surface of a Mueller-Hinton (MH) agar plate (4.0 ± 0.5 mm depth) was then evenly inoculated using a sterile cotton swab. To prevent over-inoculation, excess fluid was removed from the swab before inoculating the plate for *P. aeruginosa*. Commercially prepared antibiotic disks were applied to the agar surface within 15 min of inoculation. The plates were incubated aerobically at 35 ± 1 °C for 18 ± 2 h.

Following incubation, the diameters of the zones of inhibition were measured in millimeters. Dicks produced by HiMEdia, MH, India. The results were interpreted as Susceptible (S), Susceptible, Increased Exposure (I), or Resistant (R) by comparing the zone diameters to the clinical breakpoints specified in the EUCAST Breakpoint Tables v. 14.0. The results are summarized in Table 2.

### 2.4. Therapeutic Intervention and Outcome

The treatment regimen was multi-modal and focused on aggressive mechanical cleansing and restoration of a healthier mucosal environment, as systemic antibiotic therapy was not a viable option due to extensive resistance.

The core components of the therapy included:Nasal rest [14]: A structured “nasal rest” protocol was initiated, which involved the placement of cotton tampons to occlude the nasal passages for progressively shorter durations over a period of 17 days. This protocol aimed to reduce airflow, decrease mucosal drying, and promote healing. The patient discontinued this part of the regimen after two months due to difficulty with removing a tampon but diligently continued with the nasal irrigations.Intensive Nasal Irrigation: The patient was instructed to perform nasal irrigation at least three times per day. The irrigation solution consisted of isotonic saline mixed with a small amount of baby soap to act as a surfactant.Mechanical Debridement: Daily endoscopic removal of all visible crusts and purulent debris was performed for the first two weeks of treatment.

At the three-month follow-up, the patient reported significant subjective improvement. Nasal breathing was considerably easier, the formation of crusts was markedly reduced, and family members confirmed that the persistent, strong foul odor had completely disappeared. Endoscopic examination revealed a clean nasal cavity with minimal crusting and no signs of acute inflammation (Figure 2).

## 3. Discussion

This case report details a severe presentation of secondary atrophic rhinitis (AR), highlighting a plausible pathophysiological cascade initiated by a viral infection and culminating in a chronic, multi-drug resistant bacterial disease.

We hypothesize four-stage model, that provides a coherent pathophysiological narrative that connects the initial viral insult to the patient’s long-term clinical outcome.

**Stage 1**: Initial Viral Infection and Vascular Injury. The pathogenesis began with the SARS-CoV-2 infection in October 2021. The virus’s invasion of ACE2-expressing endothelial and epithelial cells in the sinonasal mucosa likely caused direct endothelial damage, triggering a hypercoagulable state with microthrombi formation [15,16]. This initial vascular insult, combined with acute inflammation and epithelial damage, would have impaired mucociliary clearance, setting the stage for a chronic inflammatory process [17].

**Stage 2**: Chronic Rhinosinusitis and Progressive Osteomyelitis. Following the acute infection, the patient entered a 10-month period of smoldering disease (October 2021–August 2022). The persistent microvascular occlusion led to progressive tissue ischemia. This chronic, low-grade ischemia, coupled with impaired clearance and secondary bacterial colonization, manifested as a severe chronic rhinosinusitis that inexorably progressed to involve the underlying bone, establishing a widespread osteomyelitis.

**Stage 3**: Acute Necrotizing Rhinosinusitis. The chronic ischemic and inflammatory process culminated in the acute presentation in August 2022, when the extent of tissue death became clinically overwhelming. This led to the definitive diagnosis of necrotizing rhinosinusitis with osteomyelitis, confirmed by CT findings of an “osteolytic process” and histopathology showing “purulent-necrotic detritus” and bone fragments.

**Stage 4**: Anatomical Devastation and Chronic Dysbiosis. The combination of the initial massive ischemic necrosis and the necessary radical surgical debridement of non-viable tissue in September 2022 resulted in the profound and permanent anatomical devastation that established the substrate for secondary AR. This massively widened, non-functional cavity, stripped of its ciliated epithelium and normal mucociliary clearance mechanisms, became susceptible to chronic colonization by a dysbiotic polymicrobial flora [4]. In this case, the cavity became dominated by *Pseudomonas aeruginosa* and *Staphylococcus aureus*, which perpetuate a vicious cycle of inflammation, crusting, and tissue degradation [4,6].

This progression from viral insult to anatomical destruction and subsequent chronic bacterial superinfection provides a compelling narrative that differs from more common etiologies of secondary AR, such as post-surgical changes like ENS or granulomatous diseases [10,18].

This case contributes to a growing body of evidence detailing severe, non-fungal necrotic sequelae following SARS-CoV-2 infection. The proposed vascular mechanism is strongly supported by analogous reports of post-COVID ischemic events [15,16,19]. The most direct parallels include cases of hard palate necrosis attributed to intraoperatively confirmed maxillary artery thrombosis and maxillary avascular necrosis linked to thromboembolic events in terminal arteries [20]. Another case of severe scull base osteomyelitis due to middle ear infection after COVID-19 is worth mentioning [21]. Despite different locations, the pathophysiological mechanisms are similar.

More localized events, such as full-thickness necrosis of the nasal columella and vestibule, have also been reported after even mild COVID-19, with extensive workups ruling out other infectious or vasculitic causes [22]. This pattern is not confined to the head and neck; numerous reports now document a systemic pattern of avascular necrosis in post-COVID patients, most notably in the femoral head [19]. Crucially, this has occurred in patients who received no corticosteroids, strongly implicating the virus-induced hypercoagulable state as the primary cause [23].

Furthermore, this case is distinct from typical osteomyelitis secondary to chronic rhinosinusitis (CRS). In CRS-related osteomyelitis, the process is typically a slow, contiguous erosion of bone from a primary bacterial focus [24,25]. The sheer scale of destruction, involving the entire turbinate system, septum, and lateral walls, is far more extensive than the localized osteomyelitis typically described as a complication of CRS.

The non-fungal etiology of the necrosis is a critical distinguishing feature. Invasive fungal sinusitis (IFS), particularly mucormycosis, is a well-documented post-COVID complication [26] but was definitively ruled out in this patient by two separate histopathology reports that showed no evidence of fungal invasion. Likewise, granulomatous diseases such as granulomatosis with polyangiitis (GPA), which can cause extensive nasal destruction, were considered unlikely due to the absence of characteristic granulomatous inflammation on biopsy and the lack of systemic involvement (e.g., pulmonary or renal disease) [27].

The microbiological findings are central to understanding the clinical challenges of this case. The high bacterial load (10^5^–10^6^ CFU/mL) of *P. aeruginosa* and *S. aureus*, both notorious for their potential ability to form resilient biofilms, explains the persistence of symptoms despite initial debridement [28]. Biofilms are structured communities of bacteria encased in a self-produced polymeric matrix, which adhere to surfaces and are highly resistant to host immune responses and antimicrobial agents [28]. This biofilm likely contributed significantly to the thick and tenacious crusts. Furthermore, the antimicrobial susceptibility profile of the *P. aeruginosa* isolate revealed extensive drug resistance, including resistance to all tested anti-pseudomonal beta-lactams, aminoglycosides, and even colistin, a last-resort antibiotic. This rendered conventional systemic antibiotic therapy futile and underscored the necessity of a management strategy that did not rely on antimicrobial agents.

It is critical to understand the interconnected relationship between sinonasal anatomy, mucosal physiology, and the bacterial composition of the microbiome. In a healthy state, intact nasal anatomy and a functional mucosal layer facilitate efficient mucociliary clearance. This constant mechanical removal of excess mucus and trapped bacteria is what maintains high microbiological diversity and prevents pathogenic overgrowth. When any component of this clearance system is broken, as seen in this case, it creates favorable conditions for mucus accumulation and its extensive colonization by a few dominant species. This process greatly reduces microbial diversity, leading to a state of chronic local inflammation. The primary cause of the dysbiosis in secondary AR is therefore the anatomical destruction and the failure of mucociliary clearance, rather than a primary contamination with pathogenic microorganisms like *Pseudomonas*. Consequently, therapeutic strategies aimed at microbiome modulation, such as probiotics or prebiotics, would not be expected to have a significant impact, as they require a viable mucosal habitat to be effective [6].

The role of the nasal microbiota is increasingly recognized as a key factor in the pathophysiology of chronic nasal diseases, including AR [[10], [29],]. In a healthy state, the nasal passages maintain a balanced ecosystem of commensal microbes. However, in AR, the profound anatomical and physiological changes—such as the loss of ciliated epithelium and impaired mucociliary clearance—create an environment ripe for dysbiosis [29]. This leads to a shift in the microbiome, characterized by reduced bacterial diversity and the overgrowth of opportunistic pathogens. While these microbial shifts are likely a consequence of the underlying mucosal atrophy rather than the primary cause, they establish a vicious cycle of chronic inflammation, biofilm formation, and further tissue damage. The most frequently isolated organisms in AR, as summarized in Table 2, play a significant role in perpetuating the disease state.

The microbial flora in AR is typically dysbiotic [4,6], with several key pathogens playing a significant role in perpetuating the disease state. *Klebsiella pneumoniae subsp. ozaenae* is strongly associated with primary AR and is often considered the classic causative agent of ozena [3,30,31]. *Pseudomonas aeruginosa* is a dominant pathogen, particularly in secondary AR, with one study reporting its presence in 72% of cases [32,33,34]. *Staphylococcus aureus* is also frequently isolated, especially in secondary AR, and is known for its potent biofilm-forming capabilities [35,36]. Other commonly found bacteria that contribute to the polymicrobial flora include *Proteus* spp. [37,38] and *Escherichia coli*, the latter of which is often isolated in the ‘wet’ form of secondary AR [39].

The therapeutic approach, therefore, focused on mechanical and topical strategies. Intensive daily debridement and high-volume nasal irrigation are cornerstones of AR management, aiming to remove crusts, reduce bacterial load, and hydrate the mucosa. The novel addition of baby soap to the isotonic saline irrigation was a key element of this patient’s successful treatment [29]. We hypothesize that the surfactant properties of the soap helped to disrupt the integrity of the bacterial biofilm, breaking down the protective matrix and allowing for more effective mechanical removal of bacteria and debris with irrigation. This approach directly targets the biofilm pathophysiology, which is a significant advantage in the setting of extreme antibiotic resistance. This management contrasts sharply with standard protocols for more common forms of chronic rhinosinusitis (CRS), which often rely on intranasal or systemic corticosteroids and targeted antibiotic therapy based on susceptibility data.

Another key factor in the management was the implementation of a “nasal rest” strategy. This established, though often underutilized, technique is based on the principle of mechanical obturation of the nasal vestibule. By excluding airflow, a relatively warm and humid microenvironment is created within the nasal cavity, which eliminates a primary driver of crust formation: the desiccation of mucus. Although the method can cause temporary discomfort, such as mouth dryness and difficulty speaking, its therapeutic effects are considerable. In this case, due to the severity of the disease and the aggressive microbiome, an initial intensive protocol was instituted, with the nostrils closed for three hours and then opened for one hour. As the patient’s condition improved, the period of nasal occlusion was gradually shortened to a maintenance regimen of 1–2 h, performed twice daily.

While our management focused on fundamental mechanical cleansing, other therapeutic options were considered and excluded. The use of topical antibiotics would have had little to no effect given the extreme multi-drug resistance of the *P. aeruginosa* isolate and could have further reduced microbiome diversity. More advanced surgical and regenerative therapies were also deemed inappropriate. Mucosal grafting was not feasible due to the complete lack of viable donor regions, as the usual donor sites (nasal septum, middle or inferior conchae) had been destroyed by the disease process. Similarly, the use of fillers—whether endogenous (bone, cartilage) or exogenous (hyaluronic acid-based)—was excluded due to the high risk of infection and subsequent resorption in such a heavily colonized and compromised cavity. The patient’s comorbidities, the large size of the defect, and the lack of an intact mucosal layer to cover any implants made such elective surgeries contraindicated.

The profound anatomical changes in this patient also raise the differential diagnosis of Empty Nose Syndrome (ENS), an iatrogenic condition that can follow aggressive turbinate surgery. While both secondary AR and ENS involve a widened nasal cavity and symptoms of nasal obstruction, key differences exist (Table 3). This patient’s presentation, with extensive osteomyelitis, severe ozena, and histopathological evidence of necrotizing inflammation, is more consistent with severe secondary AR than with ENS, where the primary pathology is often related to sensory dysfunction rather than overwhelming infection and tissue loss.

## 4. Conclusions

This case reports a severe, destructive form of secondary atrophic rhinitis initiated by a COVID-19-associated necrotizing infection. It highlights the potential for SARS-CoV-2 to trigger a cascade of events leading to profound and permanent sinonasal anatomical and functional impairment. The management of such cases, particularly when complicated by multi-drug resistant organisms, must shift away from a reliance on antibiotics and toward strategies focused on “nasal rest”, mechanical debridement and biofilm disruption. The use of a simple surfactant, such as baby soap, in nasal irrigation solutions may represent an effective, accessible, and low-cost adjunct for managing biofilm-associated AR. This report contributes to the growing understanding of post-COVID-19 sequelae and offers a practical management framework for a clinically devastating condition.

## Figures and Tables

**Figure 1 reports-08-00237-f001:**
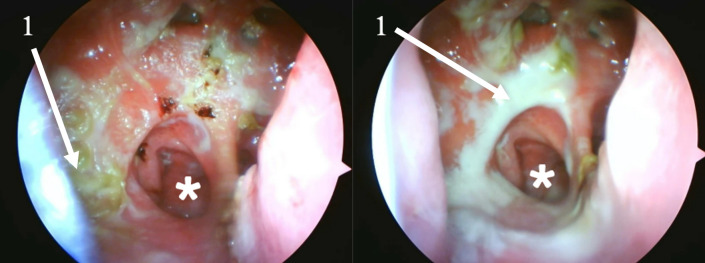
Endoscopic progression of the nasal cavity during treatment. Imaging was done using 0-degree rigid endoscope. 1—purulent discharge and crusting. *—Right choana. Large septal perforation can be observed on the right. Day 4 post-treatment initiation (**left**), showing heavy crusting and purulent discharge obscuring the underlying mucosa. Day 7 (**right**), following debridement, revealing persistent inflammation and increased purulent discharge. This transition from solid, tenacious crusts to a more liquid discharge is considered a beneficial clinical progression, as the purulent material can be effectively cleared with intensive nasal douching rather than requiring mechanical removal with forceps.

**Figure 2 reports-08-00237-f002:**
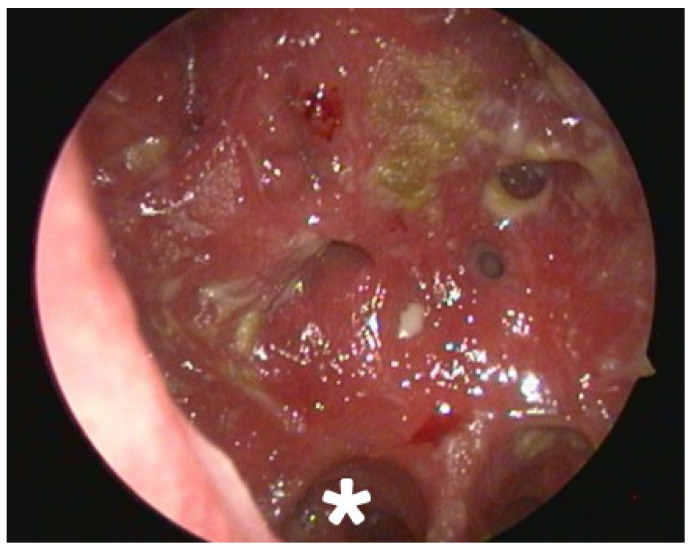
Endoscopic picture at the 3-month follow-up. *—Right choana. A relatively clean nasal cavity with significant reduction in crusting and inflammation can be observed. Imaging was done using 30-degree rigid endoscope through left nostril and different camera unit, thus contributing to different angle and colour gamma.

**Figure 3 reports-08-00237-f003:**
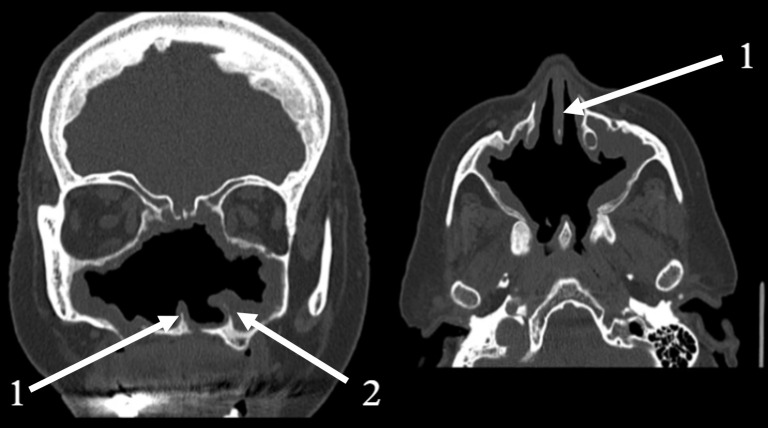
Coronal (**left**) and axial (**right**) CT scans of the paranasal sinuses. 1—remnants of nasal septum; 2—remnants of left inferior turbinate. Coronal view demonstrating the complete absence of nasal turbinates, severe resorption of the lateral nasal walls, and diffuse mucosal thickening. Axial view highlighting the extremely widened nasal cavity and a large posterior septal perforation.

**Table 1 reports-08-00237-t001:** Timeline of clinical evolution.

Date/Time Period	Key Clinical Event	Diagnostic Findings
October 2021	Acute SARS-CoV-2 Infection	Onset of persistent nasal symptoms. Continued for 10 months.
Late August 2022	Rapid worsening, onset of necrotizing rhinitis	Clinical evidence of necrosis of nasal cavity structures. Partial debrodement, histopathology: Focally necrotic tissue no fungal invasion identified. Culture: *Proteus vulgaris* (10^4^ CFU/mL), and *Enterococcus* sp. (10^3^ CFU/mL).
Late August–mid September 2022	No significant improvement	Conservative treatment with unsatisfactory results. Partial debridement mid Sep due to condition worsening
Late September 2022	Disease progression	CT scan reveals an “osteolytic process”. A major Polysinusotomy with necrosectomy is performed under general anesthesia, involving extensive removal of necrotic bone and tissue. The second histopathology report confirms the findings of necrotic tissue and bone fragments, with no evidence of fungal invasion or granylomatosis.
2022–2024	Atrophic rhinitis progression	Disease progression with gradual nasal degeneration. Treated elsewhere (occasional mechanical crust removal)
2024	Hospitalization for Unrelated Condition	Admitted for pneumonia and pulmonary embolism.
2024	Secondary atrophic rhinitis	Endoscopy, CT scan. Treatment initiation. 3-month follow-up.

**Table 2 reports-08-00237-t002:** Antimicrobial Susceptibility Profile of Nasal Isolates based on EUCAST 2024 Guidelines [7].

Antimicrobial Agent	*Pseudomonas aeruginosa* (2024)	Antimicrobial Agent	*Staphylococcus aureus* (2024)	Antimicrobial Agent	*Proteus vulgaris* (2022)
Beta-lactams		Beta-lactams		Beta-lactams	R
Ampicillin/sulbactam	-	Penicillin	-	Ampicillin	R
Ticarcillin/clavulanic acid	R	Oxacillin (Cefoxitin screen)	S	Meropenem	S
Piperacillin	R	Cefoxitin	S	Ceftazidime	S
Piperacillin/tazobactam	R	Fluoroquinolones		Ceftriaxon	I
Ceftazidime	R	Ciprofloxacin	I	Cefepime	S
Cefepime	R	Levofloxacin	I	Gentamicin	S
Aztreonam	R	Moxifloxacin	S	Amikacin	S
Imipenem	R	Aminoglycosides		Ceftazidime-avibactam	S
Meropenem	R	Amikacin	S	Cefpirome	S
Aminoglycosides		Gentamicin	S	Cefoperazone	S
Amikacin	R	Tobramycin	S		
Gentamicin	-	Macrolides/Lincosamides			
Tobramycin	R	Erythromycin	S		
Fluoroquinolones		Clindamycin	S		
Ciprofloxacin	I	Other Agents			
Levofloxacin	I	Linezolid	S		
Polymyxins		Trimethoprim/Sulfamethoxazole	S		
Colistin	R	Tetracycline	S		
Other Agents		Fusidic acid	S		
Trimethoprim/sulfamethoxazole	-	Rifampicin	S		
		Tigecycline	-		
		Chloramphenicol	-		

S, Susceptible; I, Susceptible, Increased Exposure; R, Resistant. A dash (-) indicates that EUCAST does not provide breakpoints, as the agent is considered clinically inappropriate or intrinsically ineffective against this organism and is not recommended for testing or reporting.

**Table 3 reports-08-00237-t003:** Differentiating Features of Secondary Atrophic Rhinitis and Empty Nose Syndrome (ENS).

Feature	Secondary Atrophic Rhinitis (as in this Case)	Empty Nose Syndrome (ENS)
Etiology	Severe destruction from infection, trauma, or granulomatous disease.	Iatrogenic, typically from aggressive turbinate resection.
Key Symptoms	Extensive crusting, severe foul odor (ozena), anosmia, variable nasal obstruction.	Paradoxical nasal obstruction (sensation of suffocation), nasal dryness, crusting (usually less severe than AR).
Histopathology	Squamous metaplasia with profound loss of ciliated epithelium, goblet cells, and submucosal glands. Bony resorption is common.	Ciliated epithelium and goblet cells are often preserved. Submucosal fibrosis and gland reduction may occur, but bony resorption is not a primary feature.
Sensory Perception	Anosmia due to destruction of olfactory epithelium. Sensation of airflow is lost.	Paradoxical obstruction is linked to downregulation of trigeminal thermoreceptors (e.g., TRPM8), impairing the sensation of airflow.

## Data Availability

The original contributions presented in this study are included in the article. Further inquiries can be directed to the corresponding author.

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
