# Peer review of "Severe Secondary Atrophic Rhinitis with Extensive Osteomyelitis Following COVID-19-Associated Necrotizing Rhinitis: A Case Report and Microbiological Analysis"

_reports, 2025, doi:10.3390/reports8040237_

Round 1

Reviewer 1 Report

Comments and Suggestions for Authors

The authors have presented an interesting case report of a post COVID atrophic rhinitis. It is has been worked up well.

Only one comment needs to be addressed and added to the manuscript

Lot of such destruction during COVID happened due to mucormycosis. Was that investigated and what was the result. A few sentences can be added about mucormycosis in this clinical situation.

Author Response

Reviewer: 1 (Round 1)

Dear reviewer!

Many thanks for Your time spending and efforts in reviewing the manuscript. All changes are highlighted in blue.

Point-by-point response to Comments and Suggestions for Authors

Comment:

The manuscript "Severe Secondary Atrophic Rhinitis with Extensive Osteomyelitis Following COVID-19-Associated Necrotizing Rhinitis: A  Case Report and Microbiological Analysis" reports a case of a 75 years old patient with Secondary Atrophic Rhinitis presumably related to a COVID-19 associated necrotizing rhinitis. 

Although the case described is interesting and well documented, the connection with COVID-19 which was established by the authors remains largely speculative, not only because of the timelapse between COVID-19 infection and the secondary atrophic rhinitis episode, but also because the surgical approach previously performed after necrotizing rhinitis may itself have contributed to the development of atrophic rhinitis. 

It is therefore suggested that the authors reconsider the association with COVID-19 in the manuscript, including in the conclusions, or provide further evidence of this association. 

Response:

Thank you for your comment. We have completely rewritten paragraph about initial COVID-19 infection and disease progression. Now the connection between COVID-19 and following necrotizing rhinosinusitis is represented better.

Comment: Lines 106 to 108- The information about the keywords used for literature review is not necessary in case reports and may be removed.

Response:

Thank you for your comment. Mentioned lines were removed.

Comment: Line 123- the authors should provide information about the panel of biochemical tests used for  bacterial identification

Response:

Thank you for your comment. Mentioned in the text. P. aeruginosa identity was confirmed by positive Oxidase test, non-fermentative me-tabolism on TSI, S. aureus was confirmed by positive Catalase test and Coagulase test (tube method).

Comment: Line 125- considering that the swab specimens were inoculated in solid media, how was the bacterial concentration obtained in CFU/ml? The culture was performed from the swab? Or from a transport medium?

Response:

Thank you for your comment. We have clarified the quantitative methodology.

Serial dilutions were prepared from this liquid eluate, and 0.1 mLwas plated to obtain the CFU/mL concentration.

Comment: Tables in the manuscript should have only essential information for the case report; therefore, table 2 can be removed and the information can be summarized in the text  (simply referring which are the most relevant pathogens)

Response:

Thank you for your comment. We made appropriate changes.

Comment: Line 211 - the biofilm formation was not evaluated for the identified bacteria; therefore, these pathogens may only be referred as "potentially biofilm forming agents", as this trait is not always expressed in clinically-isolated P. aeruginosa and S. aureus.

Response:

Thank you for your comment. We made appropriate changes and added a dedicated paragraph in discussion

Comment: Lines 282 and 283 contains a repeated sentence from lines 280 and 281

Response:

Thank you for the comment. Corrected.

Comment: References section - a  revision on the bacteria identification should be performed (Genus should be capitalized and italicized, species should be italicized)

Response:

Thank you for the comment. We made appropriate changes.

Sincerely,

Yulian Konechnyi

Reviewer 2 Report

Comments and Suggestions for Authors

This case report presents a rare and clinically significant manifestation of secondary atrophic rhinitis (AR) following COVID-19-associated necrotizing rhinitis. The manuscript is well-structured and clearly written, with a coherent clinical narrative supported by microbiological data and imaging. It contributes valuable insights into post-COVID sinonasal complications and antibiotic-resistant infections. However, several areas require improvement to meet the standards expected for publication in Reports (MDPI). These include refinement of scientific rigor, clarification of methodology, additional discussion on certain aspects of pathogenesis, and better integration of literature context. The paper highlights a previously unreported link between SARS-CoV-2 infection and severe secondary atrophic rhinitis with osteomyelitis, filling a gap in post-COVID ENT pathology. The clinical history, imaging, and microbiological findings are detailed, giving readers a clear understanding of disease progression and management challenges. The paper offers an accessible, low-cost therapeutic approach (mechanical debridement + surfactant irrigation) relevant for antibiotic-resistant infections. The manuscript maintains good readability and clarity for an international audience.

Several points have to be addressed by the authors (major comments): 

  • The manuscript should elaborate further on the hypothesized mechanism linking SARS-CoV-2 infection to necrotizing rhinitis and osteomyelitis. Please, discuss the vascular and immunologic effects of COVID-19 (e.g., microthrombosis, endothelial damage, immune dysregulation) in relation to local sinonasal ischemia and clarify whether this mechanism is speculative or supported by other case reports or histopathological studies.
  • Please provide more information on differential diagnosis — particularly how other causes of necrotizing rhinitis (e.g., mucormycosis, granulomatosis with polyangiitis) were ruled out.
  • Clarify whether histopathology or imaging showed any fungal invasion or vascular occlusion.
  • Mention if fungal culture or PCR testing was repeated at follow-up.
  • Please Add information about:
    • Quality control procedures for culture and AST.
    • The source of antibiotic disks and laboratory accreditation (if applicable).
    • Whether biofilm formation was directly observed or inferred.
  • The manuscript would benefit from including:
    • A table or figure summarizing the timeline of clinical evolution (infection onset, debridement, treatment start, follow-up).
    • Before-and-after CT comparisons or an annotated figure highlighting the extent of anatomical destruction.
  • Expand the discussion to include:
    • Comparison with other reported cases of post-COVID sinonasal necrosis.
    • Discussion of alternative management strategies (e.g., topical antibiotics, mucosal grafting).
    • The role of local microbiome restoration, possibly referencing probiotic or microbiota-based interventions in AR or CRS.
  • The statement about the use of Google Gemini for “language editing and literature search” should be expanded for transparency.
  • Specify that AI assistance did not generate clinical or analytical content.
  • Ensure compliance with MDPI’s AI use disclosure policy.

Minor comments

Section

Comment

Recommendation

Abstract

Too long and dense.

Shorten sentences; use structured format (Background, Case, Management, Conclusion).

Introduction

Repeats definitions from prior literature.

Condense background on primary vs secondary AR; focus more on the COVID-19 link.

Figures

Figure captions are descriptive but lack scale and orientation.

Add labels for anatomical orientation (left/right, coronal/axial) and brief legends.

Table 1

Formatting issues (uneven alignment of agents).

Standardize column widths; ensure clarity when printed.

Table 3

Excellent comparison between AR and ENS.

Add a brief explanatory note below the table summarizing its clinical relevance.

References

Several are recent and appropriate, but some lack DOI hyperlinks.

Verify all citations; ensure MDPI reference style consistency.

Language

Overall clear but some minor grammatical issues (e.g., “This report presents” vs “The present report describes”).

Perform final English proofreading before submission.

Author Response

Reviewer: 2 (Round 1)

Dear reviewer!

Many thanks for Your time spending and efforts in reviewing the manuscript. All changes are highlighted in blue.

Point-by-point response to Comments and Suggestions for Authors

Comment:

This case report presents a rare and clinically significant manifestation of secondary atrophic rhinitis (AR) following COVID-19-associated necrotizing rhinitis. The manuscript is well-structured and clearly written, with a coherent clinical narrative supported by microbiological data and imaging. It contributes valuable insights into post-COVID sinonasal complications and antibiotic-resistant infections. However, several areas require improvement to meet the standards expected for publication in Reports (MDPI). These include refinement of scientific rigor, clarification of methodology, additional discussion on certain aspects of pathogenesis, and better integration of literature context. The paper highlights a previously unreported link between SARS-CoV-2 infection and severe secondary atrophic rhinitis with osteomyelitis, filling a gap in post-COVID ENT pathology. The clinical history, imaging, and microbiological findings are detailed, giving readers a clear understanding of disease progression and management challenges. The paper offers an accessible, low-cost therapeutic approach (mechanical debridement + surfactant irrigation) relevant for antibiotic-resistant infections. The manuscript maintains good readability and clarity for an international audience.

Response:

Thank you for your high appreciation of our manuscript.

Comment: The manuscript should elaborate further on the hypothesized mechanism linking SARS-CoV-2 infection to necrotizing rhinitis and osteomyelitis. Please, discuss the vascular and immunologic effects of COVID-19 (e.g., microthrombosis, endothelial damage, immune dysregulation) in relation to local sinonasal ischemia and clarify whether this mechanism is speculative or supported by other case reports or histopathological studies.

Response:

Thank you for your comments. We have completely reworked section about link between primary COVID-19 infection and atrophic rhinosinusitis both in case presentation and discussion.

Comment: Please provide more information on differential diagnosis — particularly how other causes of necrotizing rhinitis (e.g., mucormycosis, granulomatosis with polyangiitis) were ruled out.

Response:

Thank you for your comments. Appropriate changes were made. Lines: 105-110

Comment: Clarify whether histopathology or imaging showed any fungal invasion or vascular occlusion.

Response:

Thank you for your comments. Appropriate changes were made. Lines: 88-103

Comment: Mention if fungal culture or PCR testing was repeated at follow-up.

Response:

Thank you for your comments. Appropriate changes were made. Lines: 102-103

Comment: Quality control procedures for culture and AST.

Response:

Thank you for your comments. EUCAST guidelines were used, including quality control.

Comment: The source of antibiotic disks and laboratory accreditation (if applicable).

Response:

Thank you for your comments. Dicks produced by HiMEdia, MH, India. Added info to the manuscript text.

Comment: Whether biofilm formation was directly observed or inferred.

Response:

Thank you for your comments. Appropriate changes were made. Lines: 169-178

Comment: A table or figure summarizing the timeline of clinical evolution (infection onset, debridement, treatment start, follow-up).

Response:

Thank you for this idea. We added Table 1 in order to clarify this question.

Comment:  Before-and-after CT comparisons or an annotated figure highlighting the extent of anatomical destruction

Response:

Thank you for your comments. Description of endoscopic view and CT scans were completely redone. Please check lines 222-246

Comment: Comparison with other reported cases of post-COVID sinonasal necrosis.

Response:

Thank you for your comments. Appropriate changes were made. Lines: 294-320

Comment: Discussion of alternative management strategies (e.g., topical antibiotics, mucosal grafting).

Response:

Thank you for your comments. Appropriate changes were made. Lines: 391-402

Comment: The role of local microbiome restoration, possibly referencing probiotic or microbiota-based interventions in AR or CRS.

Response:

Thank you for your comments. Appropriate changes were made. Lines: 334-347

Comment: The statement about the use of Google Gemini for “language editing and literature search” should be expanded for transparency. Specify that AI assistance did not generate clinical or analytical content. Ensure compliance with MDPI’s AI use disclosure policy.

Response:

Thank you for your comments. Appropriate changes were made. Lines: 145-153

Comment: Minor coments

Response:

Thank you for your comments. Manuscript has been heavily reorganized and upgraded.

Sincerely,

Yulian Konechnyi

Reviewer 3 Report

Comments and Suggestions for Authors

The manuscript "Severe Secondary Atrophic Rhinitis with Extensive Osteomyelitis Following COVID-19-Associated Necrotizing Rhinitis: A  Case Report and Microbiological Analysis" reports a case of a 75 years old patient with Secondary Atrophic Rhinitis presumably related to a COVID-19 associated necrotizing rhinitis. 

Although the case described is interesting and well documented, the connection with COVID-19 which was established by the authors remains largely speculative, not only because of the timelapse between COVID-19 infection and the secondary atrophic rhinitis episode, but also because the surgical approach previously performed after necrotizing rhinitis may itself have contributed to the development of atrophic rhinitis. 

It is therefore suggested that the authors reconsider the association with COVID-19 in the manuscript, including in the conclusions, or provide further evidence of this association. 

Specific comments:

  • Lines 106 to 108- The information about the keywords used for literature review is not necessary in case reports and may be removed.
  • Line 123- the authors should provide information about the panel of biochemical tests used for  bacterial identification 
  • Line 125- considering that the swab specimens were inoculated in solid media, how was the bacterial concentration obtained in CFU/ml? The culture was performed from the swab? Or from a transport medium? 
  • Tables in the manuscript should have only essential information for the case report; therefore, table 2 can be removed and the information can be summarized in the text  (simply referring which are the most relevant pathogens)
  • Line 211 - the biofilm formation was not evaluated for the identified bacteria; therefore, these pathogens may only be referred as "potentially biofilm forming agents", as this trait is not always expressed in clinically-isolated P. aeruginosa and S. aureus 
  • Lines 282 and 283 contains a repeated sentence from lines 280 and 281
  • References section - a  revision on the bacteria identification should be performed (Genus should be capitalized and italicized, species should be italicized)

Author Response

Reviewer: 3 (Round 1)

Dear reviewer!

Many thanks for Your time spending and efforts in reviewing the manuscript. All changes are highlighted in blue.

Point-by-point response to Comments and Suggestions for Authors

Comment:

The manuscript "Severe Secondary Atrophic Rhinitis with Extensive Osteomyelitis Following COVID-19-Associated Necrotizing Rhinitis: A  Case Report and Microbiological Analysis" reports a case of a 75 years old patient with Secondary Atrophic Rhinitis presumably related to a COVID-19 associated necrotizing rhinitis. 

Although the case described is interesting and well documented, the connection with COVID-19 which was established by the authors remains largely speculative, not only because of the timelapse between COVID-19 infection and the secondary atrophic rhinitis episode, but also because the surgical approach previously performed after necrotizing rhinitis may itself have contributed to the development of atrophic rhinitis. 

It is therefore suggested that the authors reconsider the association with COVID-19 in the manuscript, including in the conclusions, or provide further evidence of this association. 

Response:

Thank you for your comment. We have completely rewritten paragraph about initial COVID-19 infection and disease progression. Now the connection between COVID-19 and following necrotizing rhinosinusitis is represented better.

Comment: Lines 106 to 108- The information about the keywords used for literature review is not necessary in case reports and may be removed.

Response:

Thank you for your comment. Mentioned lines were removed.

Comment: Line 123- the authors should provide information about the panel of biochemical tests used for  bacterial identification

Response:

Thank you for your comment. Mentioned in the text. P. aeruginosa identity was confirmed by positive Oxidase test, non-fermentative me-tabolism on TSI, S. aureus was confirmed by positive Catalase test and Coagulase test (tube method).

Comment: Line 125- considering that the swab specimens were inoculated in solid media, how was the bacterial concentration obtained in CFU/ml? The culture was performed from the swab? Or from a transport medium?

Response:

Thank you for your comment. We have clarified the quantitative methodology.

Serial dilutions were prepared from this liquid eluate, and 0.1 mLwas plated to obtain the CFU/mL concentration.

Comment: Tables in the manuscript should have only essential information for the case report; therefore, table 2 can be removed and the information can be summarized in the text  (simply referring which are the most relevant pathogens)

Response:

Thank you for your comment. We made appropriate changes.

Comment: Line 211 - the biofilm formation was not evaluated for the identified bacteria; therefore, these pathogens may only be referred as "potentially biofilm forming agents", as this trait is not always expressed in clinically-isolated P. aeruginosa and S. aureus.

Response:

Thank you for your comment. We made appropriate changes and added a dedicated paragraph in discussion

Comment: Lines 282 and 283 contains a repeated sentence from lines 280 and 281

Response:

Thank you for the comment. Corrected.

Comment: References section - a  revision on the bacteria identification should be performed (Genus should be capitalized and italicized, species should be italicized)

Response:

Thank you for the comment. We made appropriate changes.

Sincerely,

Yulian Konechnyi

Round 2

Reviewer 2 Report

Comments and Suggestions for Authors

I have no further comments.